# The Genetic and Molecular Basis of Developmental Language Disorder: A Review

**DOI:** 10.3390/children9050586

**Published:** 2022-04-20

**Authors:** Hayley S. Mountford, Ruth Braden, Dianne F. Newbury, Angela T. Morgan

**Affiliations:** 1Department of Biological and Medical Sciences, Oxford Brookes University, Oxford OX3 0BP, UK; hmountford@brookes.ac.uk (H.S.M.); diannenewbury@brookes.ac.uk (D.F.N.); 2Murdoch Children’s Research Institute, Royal Children’s Hospital, Melbourne 3052, Australia; ruth.braden@mcri.edu.au

**Keywords:** language disorder, apraxia of speech, CAS, DLD, genetics, specific language impairment, SLI, neurodevelopment, heritability, epigenetics

## Abstract

Language disorders are highly heritable and are influenced by complex interactions between genetic and environmental factors. Despite more than twenty years of research, we still lack critical understanding of the biological underpinnings of language. This review provides an overview of the genetic landscape of developmental language disorders (DLD), with an emphasis on the importance of defining the specific features (the phenotype) of DLD to inform gene discovery. We review the specific phenotype of DLD in the genetic literature, and the influence of historic variation in diagnostic inclusion criteria on researchers’ ability to compare and replicate genotype–phenotype studies. This review provides an overview of the recently identified gene pathways in populations with DLD and explores current state-of-the-art approaches to genetic analysis based on the hypothesised architecture of DLD. We will show how recent global efforts to unify diagnostic criteria have vastly increased sample size and allow for large multi-cohort metanalyses, leading the identification of a growing number of contributory loci. We emphasise the important role of estimating the genetic architecture of DLD to decipher underlying genetic associations. Finally, we explore the potential for epigenetics and environmental interactions to further unravel the biological basis of language disorders.

## 1. Introduction

Our ability to learn and utilise language early in our lives is often considered a central feature of human evolution. Children develop receptive and expressive language ability early in life, learning and applying seemingly cryptic morphological, phonological, and syntactic rules, alongside semantic and pragmatic meanings. Most children progress their language development in a relatively structured way throughout their schooling; however, some children struggle to keep up with their classmates. More than 7% of UK school-age children meet criteria for developmental language disorder (DLD) [1] defined as language ability substantially below their peers that affects their everyday function and cannot be explained by another medical diagnosis. Children with substantial language difficulties that impact everyday life have a language disorder. If that language disorder is associated with a medical condition, then it is termed “Language Disorder associated with X medical condition”. If no explanatory medical condition is present, then the child meets the criteria for DLD, where a secondary descriptor of a co-occurring condition or area of particular difficulty (e.g., phonological) is included [2]. In real terms, there will be at least two children who meet the criteria for DLD in every classroom.

Language disorders have been shown to have a profound life-long impact: deficits in communication disrupt social, emotional, and educational development. These conditions increase the risk of behavioural disorders, and ultimately unemployment and mental health issues in adulthood [3]. Despite language disorders carrying an extremely high social and economic burden, we understand little of the biology which underlies them. While there have been more than thirty genes identified that play a role in speech and/or language disorders (reviewed in this paper), this still only explains the underlying genetic cause for a fraction of DLD cases. Limitations in phenotyping are one of the factors contributing to the impoverished genetic evidence base for language disorders; exacerbated by a lack of international diagnostic consensus guidelines and gold-standard tests, leading to heterogeneity between research studies and making direct comparison or replication of results challenging. Importantly, this heterogeneity has severely restricted sample sizes, further limiting replication of genetic findings. As such, new discoveries in the genetic basis of language disorders have lagged staggeringly behind that of other neurodevelopmental disorders such as ASD, intellectual disability and attention-deficit/hyperactivity disorder (ADHD).

In this review, we will explore the current genetic landscape of language disorders and explain how the boundaries between monogenetic and complex genetic disorders are beginning to blur. We will show how recent global efforts to unify diagnostic criteria have vastly increased sample size and allow for large multi-cohort metanalyses, leading the identification of a growing number of contributory loci.

## 2. The DLD Phenotype

Historical terminology for idiopathic language disorder has included specific language impairment (SLI), or more recently developmental language disorders (DLDs) [2,4]. The term SLI was historically used to describe deficits in language ability relative to non-verbal intelligence in children with idiopathic language disorder. There has never been a formalised international consensus on the specific criteria of language deficit or non-verbal intelligence required for a diagnosis of SLI.

Historically, the SLI literature typically used language testing scores greater than 1.25 standard deviations below the mean and Performance IQ scores of 85 or higher as the cut-off for an SLI diagnosis [5,6,7] In contrast, the ICD-10 [8] defined a diagnosis of SLI where language abilities were more than 2.0 standard deviations below the mean, and at least 1.0 standard deviation lower than non-verbal intelligence. However, in practice, a vast range of cut-offs have been used in the literature [9]. To address this lack of consensus in the literature and promote better standards in terminology and diagnostic criteria, a Delphi study led by the CATALISE Consortium recommended the umbrella term *developmental language disorder* (DLD) as the terminology for idiopathic language impairment [2,4].

Importantly, the CATALISE study re-conceptualised the DLD label to acknowledge co-occurring medical conditions, which the term SLI did not. Co-occurring conditions are extremely common and a frequent feature of neurodevelopmental disorders. Further, there is high co-occurrence of DLD with other developmental conditions, including speech sound disorders (11–77%—[10]; 40.8%—[11]) and literacy disorders such as dyslexia (17–29%—[12]), suggestive of shared aetiologies between these conditions.

Children with DLD characteristically present with a history of delayed early communication milestones such as age of first word acquisition, production of two-word combinations and sentence generation. As these children reach primary school age, DLD manifests as low scores relative to peers, on measures of receptive and expressive language, including across vocabulary, grammar, and pragmatic abilities and literacy in most instances [13].

Both genetic and environmental risk and protective factors are thought to influence a child’s acquisition and trajectory of language development. For example, maternal education and socioeconomic status are predictors for language outcomes [14]. This interaction between genetics and environment therefore leads to a high degree of heterogeneity amongst DLD populations.

## 3. The Genetics of Language Disorders

Early work by Bishop et al. (1995) [15] first demonstrated that language disorders (SLI) had a strong inherited component. They found that monozygotic twins showed a high degree of concordance; in instances where one twin met the criteria for SLI, almost 100% of the co-twins also did. Later studies contrasted this finding, estimating the heritability of DLD/SLI at approximately 18% [16], and showing that the case for high heritability may not be so clear cut. Bishop and Hayiou-Thomas (2008) [17] later went on to demonstrate that the way in which the cohort was ascertained had a huge effect on heritability; where twins were ascertained clinically, heritability was extremely high (0.97), whereas it was near zero for twins who were ascertained through population-based screening. The picture for language disorders is arguably more complex than that seen in other neurodevelopmental disorders. For example, autism spectrum disorder (ASD) and attention-deficit/hyperactivity disorder (ADHD) are thought to be approximately 80% [18,19], although it is important to note that estimates vary between studies, measures and age.

Similarly, Stromswold (1998) [20] found that the risk of language disorder increased if a first degree relative also has a diagnosis. Dale et al. (1998) [21] and Spinath et al. (2004) [22] both found that language disorders are more highly heritable than general language ability, suggesting that the role of genetics may differ between disorder and language ability in general.

Despite the strong evidence for the influence of genetics, the specific changes to the genetic code and how these changes interact with the environment are poorly understood. The genetics of language disorders falls into two camps: monogenic disorders, where a single change in the DNA is sufficient to cause disorder, and complex disorders, where many genetic changes combine to contribute to an overall susceptibility or risk which is further influenced by the environment. It is important to note that the boundaries between these two camps are increasingly blurred as we begin to better understand the role of genetic variants, and how environmental conditions can influence them.

Traditionally, monogenic and complex genetic methods have been distinct from each other and utilise separate methodologies for discovering the genes that are involved. Monogenic disorders have traditionally been studied within families, whereas if a disorder is thought to be genetically complex, then genetic association methods are used. The following sections provide a review of the key findings through monogenic and complex approaches in the language disorder field.

## 4. Monogenic Speech and Language Disorders

The most studied and best understood genetic causes of language disorders are those caused by monogenic inheritance of rare variants. Monogenic (meaning “one gene”) inheritance is when a specific disorder arises from either recessive (two copies of the variant such that both gene copies are affected) or dominant inheritance (one copy of the variant in a gene where two full working copies are necessary for function). There are examples of both recessive and dominant language disorders, although these are individually very rare, often only a handful of people in the world are affected by each specific genetic disorder, and often by different variants in the same gene. This is because monogenic variants are extremely rare in the general population; often fewer than one in a hundred thousand people may carry a particular variant, and some variants are novel, meaning they have never been observed in the population before. These monogenic variants will have a negative effect on the protein for which they encode, and this in turn leads to a deficit in the cell which results in the specific disorder.

One of the best-known examples of a monogenic inheritance pattern is the gene, *FOXP2*, where a rare and detrimental variant led to childhood apraxia of speech (CAS) in the KE family [23]. CAS is described as a deficit in the motor programming and planning necessary to perform the movements required for speech distinct language disorder and is considered a specific and separate disorder under the DLD umbrella. Variants in *FOXP2* are well characterised (for review see [24]) and often show dominant inheritance, meaning that carriers are affected and non-carriers are unaffected. Individuals with reduced levels of the FOXP2 protein have receptive and expressive language difficulties with delayed and unintelligible speech but typical or low-average non-verbal intelligence [24]. Despite the speech-specific motor difficulties, affected individuals have intact gross- and fine-motor skills. *FOXP2* variants are extremely rare and only account for approximately 2% of CAS cases [25]. Interestingly, they are not thought to contribute to other forms of language disorder [26,27].

The discovery of *FOXP2* as the first gene implicated in speech and language disorders provided a window into the biology of speech and language. *FOXP2*, a transcription factor, is involved in the downstream control of many other genes important for a huge range of biological processes. Examination of these downstream interactions allowed for the identification of targets of *FOXP2*, such as *CNTNAP2* and *FOXP1* [28,29,30,31,32,33]. 

More recent advances in sequencing technology and the availability of population-level variant data have made significant changes to the methodological approaches that are used to identify causative genes. Exome or whole-genome sequencing allows for all genes to be examined at one time, rather than on a candidate gene basis, and most monogenic research now utilises this method. One such example is Chen et al. (2017) [34], who performed whole-exome sequencing on 43 unrelated individuals diagnosed with severe forms of SLI. Chen et al. found three variants in the genes *ERC1*, *GRIN2A* and *SRPX2* that fully explained the language difficulties in the carriers and identified several new candidate genes (Table 1) that were previously implicated in other neurodevelopmental disorders.

At this time, further genetic sequencing studies of SLI, or rather DLD, remain elusive, although further progress has been made in the field of severe speech disorder and namely CAS, which typically co-occurs with language impairment. Eising et al. (2019) [26] applied a *de novo* paradigm to 19 individuals with CAS. A *de novo* paradigm is where an affected child (with unaffected parents) carries a spontaneously occurring dominant variant that is not inherited from either parent. They detected rare de novo variants in the *CHD3*, *SETD1A* and *WDR5* genes. The importance of *CHD3* has since been confirmed in a large study of 34 individuals who carried variants in *CHD3* [40]. Loss-of-function variants in this gene lead to speech and language deficits accompanied by macrocephaly often in the presence of severe neurodevelopmental difficulties, named Snijders Blok-Campeau syndrome [40]. CHD3 has previously been shown to interact with the FOXP2 protein, showing that shared molecular pathways may exist between these genes [58]. As in the case of *FOXP2*/*CNTNAP2*/*FOXP1*, this suggests that the investigation of shared pathways denotes a worthwhile approach to gene identification. Indeed, Eising et al. (2019) [26] used shared pathways in brain development to identify an additional five candidate CAS genes: *KAT6A*, *SETBP1*, *ZFHX4*, *TNRC6B* and *MKL2*.

More recently, Hildebrand et al. (2020) [39] examined the DNA of 33 children with CAS (including one twin pair) and were able to identify a causative variant in eleven participants in ten genes *CDK13*, *EBF3*, *GNAO1*, *GNB1*, *DDX3X*, *MEIS2*, *POGZ*, *SETBP1*, *UPF2*, and *ZNF142*, and a 5q14.3q21.1 deletion. This implies that as many as one in three children with CAS carry a pathogenic variant causative for their speech disorder. *SETBP1* had been previously reported in an isolated Russian population with a high incidence of SLI [54]. This replication of findings across multiple studies is vital for growing a body of evidence for the role of a gene in language disorders, and is especially important when variants are rare and few cases exist. Isolated populations, such as that of Kornilov et al. (2016) [54], can be useful to identify founder effects in related individuals where there is a higher than expected occurrence of a particular phenotype. Most recently, Morgan et al. (2021) [55] utilised a reverse phenotyping approach in a cohort of children with variants in *SETBP1*. They confirmed the relevance of this gene for speech and language, demonstrating that language was differentially affected compared to other skills. Whilst the work of Eising et al. (2019) [26] and Hildebrand et al. (2020) [39] ascertained children on the basis of CAS, most children with CAS do share co-occurring expressive and/or receptive language difficulties and hence the work is still of relevance to the current review.

The identification of *NFXL1* found in individuals with DLD in the Robinson Crusoe Island population is another excellent example of using family structures to discover causative variants [53]. Both of these approaches exploit population history to select sets of individuals where genetic variants were most likely to have a large effect and be similar between affected individuals.

Finally, most recently, Andres et al. (2021) [37] reported *BUD13* as a large-effect-size rare variant shared by multiple unrelated Canadian families in which at least one member met the criteria for SLI. Table 1 summarises the known genes implicated in monogenic DLD.

Many other genes have been identified that are involved in speech and language, but either as part of a broader syndrome or present as a secondary non-idiopathic phenotype. One such example of this is a recent study which identified 42 individuals with causative variants in the gene *SATB1*, where individuals presented with a range of neurological symptoms including intellectual disability, developmental delay and motor difficulties [59]. Although the described *SATB1* syndrome is primarily considered as intellectual disability, language difficulties were observed in 89% of cases [59].

### Copy Number Variants (CNVs)

Copy number variants (CNVs) are deletions or duplications of regions of genetic material ranging from a few hundred bases through to entire arms of chromosomes. We each carry many CNVs; some are inherited from our parents, and some are de novo, some have little effect on our biology and are tolerated, whereas others can be disease causing [60]. As such, the effect of individual CNVs can be difficult to determine and often depends on the genes that are affected. Some CNVs are extremely detrimental and lead to clear genetic conditions. Some of these micro-deletion/-duplication syndromes affect global neurodevelopment, while others have been linked to language disorders. Deletions of chromosome 16p11 have been associated with a penetrant form of CAS [61,62,63].

CNVs spanning the *FOXP2* gene invariably result in the CAS phenotype because they disrupt the function of this critical gene in the same way as the single variants described above [64,65,66]. Other CNVs have been reported in single cases and led to the identification of new candidate genes *BCL11A* [36], *ERC1* [42] and *SEMA6D* [67], all of which were later validated through sequencing studies [34,38]. Morgan et al. (2017) [55] showed that patients with Koolen de Vries Syndrome caused by either a 17q21.31 microdeletion or variants in the *KANSL1* typically present with CAS and dysarthria. Mapping of deletions and duplications is another traditional method of gene mapping that has been applied to DLD.

In the same way as rare variants, CNVs have been linked to developmental disorders more widely [68], including autism [69], intellectual disability [70] and ADHD [71]. The overall burden of CNVs, meaning the number of CNVs an individual carries and the total size of the genome covered by CNVs, has been shown to play a role in DLD. Simpson et al. (2015) [72] detected increased CNV burden in cases and their unaffected relatives but concluded that the most important factor was which genes were disrupted by the CNVs. Kalnak et al. (2018) [73] found that individuals with DLD harboured more and larger rare CNVs than compared to typically developing controls.

## 5. Common Genetic Model

Monogenic causes of language disorders remain comparatively rare, and do not fully account for the DLD prevalence rate of >7% [1]. It is widely accepted that common risk variants confer a genetic susceptibility for DLDs. Termed ‘complex genetic model’, each variant contributes incrementally to an overall level of risk of developing a language disorder. Studies to identify these risk variants within a complex genetic model fall into two main approaches: linkage studies and genome-wide association studies (GWASs).

Linkage studies identify shared chromosomal regions between individuals (sometimes related) more similar at the phenotypic level. GWAS analyses involve a higher resolution of genetic markers to detect variants which are more common in affected cases than in controls. Both approaches make the same assumption that a small number of shared variants are contributing to the disease, and that the individuals in the study are genetically similar to each other, meaning that they are of the same ethnicity.

### 5.1. Linkage Analyses

Using genetic linkage studies to identify regions of the genome shared between affected individuals were the linchpin of neurodevelopmental genetics in the 2000s. As the principal method to detect common variants, they were used to generate many important findings to elucidate the genetics of complex disease. Linkage studies were particularly well suited to detect common variants with a moderate or large effect size and present in more than 10% of population. The Specific Language Impairment Consortium (SLIC) found regions strongly associated with SLI on 16q24 (SLI1) and 19q13 (SLI2) [74] (Table 2). Fine-mapping of these regions implicated two specific genes; C-mad inducing protein (*CMIP*) and calcium-transporting ATPase type 2C member 2 (*ATP2C2*) [75]. Both *CMIP* and *ATP2C2* were found to contain common risk variants with a moderate effect size and have additional evidence through the identification of monogenic cases of language disorder (see Table 2). Newbury et al. (2009) [75] discovered that *CMIP* was associated with language, reading and spelling in both the SLI cohort and in the general population. This may suggest a contribution to phonological language skills in language ability more generally. In contrast, *ATP2C2* was associated with phonological memory in the SLI cohort, but only showed association within the language-impaired group in the general population, suggesting a possible role specifically in individuals with language disorders. Recently, Martinelli et al. (2021) [76] characterised the functional effects of a rare variant in *ATP2C2* and its role in language disorders.

Bartlett et al. (2002) [85] showed a strong association to chromosomal region 13q21 (SLI3) on reading specific measures, and more modestly to regions 2p22 and 17q23 using more general measures of delayed language. In this case, they utilised a family-based linkage approach using five large Canadian families where several family members had a diagnosis of SLI. Evans et al. (2015) [86] identified two associated regions using 147 pairs of siblings, where at least one had an SLI diagnosis. They reported two regions associated with phonological memory, 10q23.33 and 13q33.3.

In many of these examples, the exact genes and contributory variants remain unelucidated. This is a particularly difficult issue with linkage studies, which tend to identify very large regions containing many hundreds of genes, making fine mapping difficult.

As cohort sizes have increased, and genomic data at a population level becomes more available, the linkage study has been replaced by the genome-wide association study except in large families or highly related populations. GWASs provide a much higher resolution of variants and allow for more efficient finer mapping of contributory variants. For example, Andres et al. (2019) [87] used linkage to identify a region on chromosome 2q associated with SLI in fourteen consanguineous Pakistani families, totalling 156 individuals.

Linkage studies are difficult to replicate in other populations, and there is rarely overlap between studies [88]. In unrelated individuals, we would expect many variants to contribute to risk, whereas a linkage study assumes that a small number of variants will have a large effect size. As such, the power to detect the genetic signal is insufficient. To do that, we need larger sample sizes and the GWAS.

### 5.2. Genome-Wide Association Studies

The current methodology for detecting genomic variants associated with a complex condition is to perform a genome-wide association study (GWAS). GWAS uses advances in genetic marker technology to simultaneously assess more than 4 million sites of known common variation across the entire genome, providing higher resolution. In addition to more variants, the number of individuals has increased into the tens or hundreds of thousands. A pivotal study in the broader field of psychiatric genetics identified more than 100 regions associated with schizophrenia using 37,000 cases and 113,000 unaffected controls [89]. Studies focused on DLD and language-related phenotypes are only very recently beginning to achieve sample size on this scale (Table 2). A number of GWASs have been performed on SLI/DLD and related phenotypes, and identified genetic regions are summarised in Table 2. As with the linkage studies described in the previous section, there is little consistency between the genomic regions found to be associated between studies. This can be partially explained by differences in phenotyping used between studies, exacerbated by the lack of robust consensus criteria for diagnosing DLDs. Secondly, the genetic aetiology of DLDs are such that it likely involves many contributing variants across many different genes (and environmental factors) each of a small effect size.

Very recently, two research studies have presented meta-analyses in which multiple different GWAS cohorts are combined into one large study [83,84]. Meta-GWAS is a method whereby the GWAS summary statistics from more modest cohort sizes can be pooled together to increase power and is a cost-efficient means of gene identification. Eising et al. (2021) [83] utilised 22 different cohorts and five measures: word reading, non-word reading, spelling, phoneme awareness and non-word spelling. They found that word reading associated with a variant (rs11208009) using a subset of 19 cohorts and 33,959 individuals. The variant lies outside of a genic region, but is located near to (and in linkage disequilibrium with) three potential candidate genes: *DOCK7*, *ANGPTL3*, and *USP1*. They went on to show that both reading and language traits have a genetic basis that is largely separate to that of Performance IQ.

Even larger still, Doust et al. (2021) [84] utilised population and genetic data from 23 and Me, totalling 51,800 adults who self-reported that they had dyslexia and over a million controls without dyslexia. The authors identified 42 individual genomic regions that associated with diagnosis of dyslexia. Of these 42, 17 had been previously reported as associated with either education attainment or cognitive ability, and 25 were novel [84]. A total of 12 of the 25 novel regions went on to be independently replicated in separate cohorts. It is important to note the trade off in these studies between the sample size and phenotyping information, which was a yes/no question and self-reported.

Both Eising et al. (2021) [83] and Doust et al. (2021) [84] represent substantial leaps forward in understanding the genetic contribution to DLDs and sample size is now large enough to detect some of the missing heritability of language disorders. As the list of candidate genes grows, so does our knowledge of the biology of DLD risk. Polygenic profiles, in which an overall risk score is generated for each risk allele associated with a phenotype, can be generated which capture the genetic differences and similarities between neurodevelopmental disorders. Early studies suggest that this is a promising area of research. Shared genetic effects have been shown to exist between cognitive ability, educational attainment, language development and psychosocial outcomes; however, this pilot study was based on very small sample sizes [90]. As polygenic profiles are updated using summary statistics from increasingly large GWASs, they become more sensitive and specific, allowing for improving inference accuracy. For example, the first profiling of educational attainment explained 2% of variance [91] while more recent scores explain 13% [92]. Polygenic methods are being developed in language disorders [90] and dyslexia [93]. Polygenic risk scores of clinical conditions indicate that polygenic profiles can be informative for the extremes of the population (who carry a high burden of risk or protective variants). So even if they are not useful for capturing individual variation in the middle of the distribution at the extremes, they can be clinically meaningful.

## 6. Missing Heritability

The two models of genetic inheritance, common and monogenic models, can only partially explain the genetics of language disorders. This is referred to as the ‘missing heritability’, reflecting the gap in knowledge of how genetic differences drive language disorders. There are three areas which may be particularly fruitful in unravelling the biology of language disorders.

Gene–gene interaction, also known as epistasis, is when two independent variants interact with each other in combination to cause a particular phenotype. They are thought of as a modifier or a “second hit”. Gene–gene interactions are known to play a role in other neurodevelopmental conditions [62,94,95,96,97]. Although there are no studies, to our knowledge, it is likely that gene–gene interactions will play a role in risk and symptom variability [95]. Several examples have been identified in dyslexia and reading-related traits [98,99], further supporting the potential of gene–gene interactions as a potential contributory mechanism.

Gene–environment interaction is when a genetic variant interacts with an environmental influence and results in a particular phenotype. There are no robust examples of this in language disorders, to our knowledge, but is it very likely to play a role. Again, the field of dyslexia provides an example of gene–environment interaction between risk factors such as socioeconomic status, maternal smoking and low birth weight and the gene *DYX1C1* [100]. Ultimately, this line of investigation may provide evidence of gene–environment interactions which differ between tissue types and throughout the developmental course.

Epigenetics is essentially the regulation of genes; turning gene expression on or off in response to the environment. Two of the most well characterised mechanisms are methylation of the DNA and modification of histones. Both have been implicated in neurodegenerative conditions such as Alzheimer’s disease [101] and Parkinson’s disease [102]. While epigenetics has been proposed by a number of groups to play a role in DLDs [103,104,105], no studies have successfully shown a specific epigenetic association. In terms of language ability in the more general adult population, Marioni et al. (2018) [106] identified a specific methylation marker within the gene *INPP5A* that was associated with verbal fluency, logical memory and vocabulary although the authors are careful to point out that these correlations should be cautiously interpreted.

Similarly, evidence from other fields suggests that some prenatal epigenetic changes persist throughout life [107] and that some of these changes may be relevant to early brain development [108]. Collectively, these three areas indicate strong potential for future research as specific genes and pathways are identified.

## 7. Phenotyping in Genetic Studies of Developmental Language Disorder

The historical changes in nomenclature and a lack of consensus over diagnostic criteria for DLD have led to varied phenotyping approaches in research studies examining the genetic architecture of this condition. A summary of the variation in phenotypic inclusion criteria for key genetic studies of SLI/DLD is presented in Table 3. This variation makes comparison and replication of results across genetic studies challenging. Further, it hinders the amalgamation of cohorts to increase sample sizes to lead to adequately powered meta-analyses. Clearly defined diagnostic criteria for language disorders is a critical aspect of methodological design for investigative genetic studies of DLD. The following section details genetic analysis approaches used in the field to date.

## 8. Conclusions

This review has provided an overview of DLD and its genetic aetiology. Recent advances in genetic association studies, predominantly achieved through substantial increases in sample size and meta-analysis, are the first indications we are making headway into understanding the genetic architecture of DLDs. Recent advances in genetic analysis have allowed for more in-depth research and the discovery of associated gene pathways. Past studies have been based on varied inclusion criteria, reflecting the historical lack of a consensus definition or classification system for SLI/DLD. The importance of consistent, fine-grained phenotyping in genetic studies of these populations going forward is thus important in order that studies may be replicated. Sophisticated deep phenotyping of language, speech and cognitive abilities will be critical for understanding the genotype–phenotype interactions of candidate genes. Epigenetic, gene–gene and gene–environment influences are also extremely likely to contribute to the phenotypic variation observed and will be an important area of future research. Ultimately, improvements in understanding the biology of DLD are of critical importance in the clinical setting for timely diagnoses and genetic counselling, while allowing for the future development of targeted therapies and improved long-term outcomes for individuals with DLD.

## Figures and Tables

**Table 1 children-09-00586-t001:** Summary of monogenic causes of idiopathic speech (i.e., CAS) and language disorders.

Gene	Type	Phenotype	Authors
*ATP2C2*	del	DLD	Smith et al. (2015) [35]
*BCL11A*	del	CAS	Peter et al. (2014) [36]
*BUD13*		SLI	Andres et al. (2022) [37], Soblet et al. (2018) [38]
*CDK13*		CAS	Hildebrand et al. (2020) [39]
*CHD3*		DLD/CAS, Snijders Blok-Campeau syndrome	Eising et al. (2018) [26], Snijders Blok et al. (2018) [40]
*CNTNAP2*		DLD	Worthey et al. (2013) [41]
*DDX3X*		CAS	Hildebrand et al. (2020) [39]
*EBF3*		CAS	Hildebrand et al. (2020) [39]
*ERC1*		CAS	Chen et al. (2017) [34], Thevenon et al. (2013) [42]
*FOXP1*		CAS	Hamdan et al. (2010) [29], Horn et al. (2010) [30], Sollis et al. (2015) [33], Srivastava et al. (2014) [43], Le Fevre et al. (2013) [44]
*FOXP2*	Includes del	CAS	Lai et al. (2001) [23], MacDermot et al. (2005) [25], Reuter et al. (2017) [45], Moralli et al. (2015) [46], Turner et al. (2013) [47], Tomblin et al. (2009) [48]
*GNAO1*		CAS	Hildebrand et al. (2020) [39]
*GNB1*		CAS	Hildebrand et al. (2020) [39]
*GRIN2A*		DLD and epilepsy, with or without intellectual disability	Chen et al. (2017) [34], Carvill et al. (2013) [49], Endele et al. (2010) [50], Turner et al. (2015) [51]
*KAT6A*		CAS	Eising et al. (2018) [26]
*KANSL1*	del	CAS	Morgan et al. (2017) [52]
*MEIS2*		CAS	Hildebrand et al. (2020) [39]
*NFXL1*		DLD	Villanueva et al. (2015) [53]
*POGZ*		CAS	Hildebrand et al. (2020) [39]
*SETBP1*		CAS	Eising et al. (2018) [26], Hildebrand et al. (2020) [39], Kornilov et al. (2016) [54], Morgan et al. (2021) [55]
*SETD1A*		CAS	Eising et al. (2018) [26]
*SRPX2*		DLD with rolandic seizures	Chen et al. (2017) [14]
*TM4SF20*	del	DLD	Wiszniewski et al. (2013) [56]
*TNRC6B*		CAS	Eising et al. (2018) [26]
*UPF2*		CAS	Hildebrand et al. (2020) [39]
*WDR5*		CAS	Hildebrand et al. (2020) [39]
*ZFHX4*		CAS	Hildebrand et al. (2020) [39]
*ZNF142*		CAS	Hildebrand et al. (2020) [39]
*ZNF277*	del	SLI	Ceroni et al. (2014) [57]

**Table 2 children-09-00586-t002:** Summary of findings from genome-wide association studies of language disorders.

Study	Sample No.	Cohort Type	Chr. Assoc.
Luciano et al. (2013) [77]	~6500	Population	21
Eicher et al. (2013) [78]	~170	Selected reading and language impaired	3, 4, 13
Nudel et al. (2014) [79]	~250	Selected (parent of origin)	5, 14
St Pourcain et al. (2014) [80]	~10,000	Population	3
Gialluisi et al. (2014) [81]	~1800	Selected reading and language impaired	7, 21
Harlaar et al. (2014) [82]	~2000	Population	2, 10
Kornilov et al. (2016) [54]	~400	Isolated population	9, 21
Eising et al. (2021) [83]	33,959	Selected and population Meta-analysis using 19 cohorts	1
Doust et al. (2021) [84]	51,800 dyslexia cases, 1,087,070 controls	Selected and population Meta-analysis using binary case/control self-reported measure of dyslexia	1, 2, 3, 6, 7, 11, 17, X

**Table 3 children-09-00586-t003:** Summary of phenotypic inclusion criteria for key studies of DLD/SLI.

Authors	Study	Diagnostic Term	Inclusion Criteria	Exclusion Criteria
Bishop et al. (1995) [15]	Twin study	SLI	Language: DSR-III-R criteria: SS ≤ 80 on language measure; significant impairment on ≥1 of 4 language measures Cognition: Discrepancy of ≥20 points between non-verbal IQ and language measure	Mental retardation; ASD; SNHL; structural abnormality of articulators; serious visual impairment; medical syndrome; EAL status
Bartlett et al. (2002) [85]	Linkage study	SLI	Language: Spoken Language Quotient (SLQ) SS ≤ 85 Cognition: Performance IQ ≥ 80 + Performance IQ ≥ SLQ	Hearing impairment; motor impairments or oral structural deviations affecting speech or non-speech movement of the articulators; diagnosis of ASD, schizophrenia, psychoses, or neurological disorder
Falcaro et al. (2008) [109]	Linkage	SLI	Language: Language SS ≤ 1SD at 1 time point during longitudinal study + Attending language units in United Kingdom Cognition: Performance IQ ≥ 80	Sensorineural hearing loss; EAL status; Medical condition likely to affect language; ASD diagnosis
Newbury et al. (2009) [75]	Linkage study	SLI	Language: * CELF-R expressive or receptive SS ≥ 1.5SD below normative mean Cognition: * Performance IQ ≥ 80	* MZ twinning, chronic illness requiring multiple hospital visits or admissions, deafness, an ICD-10/DSM-IV diagnosis of childhood autism, EAL, care provision by local authorities, and known neurological disorders
Villanueva et al. (2011) [110]	GWAS	SLI	Language: Phonology, expressive and receptive morphosyntax SS > 2 SD below population mean on Test para Evaluar Procesos de Simplificacio’n Fonolo´ gica (TEPROSIF) or Toronto Spanish Grammar Exploratory test Cognition: Performance IQ > 80th percentile	HI; oral motor or structural; ASD, emotional difficulties, or neurological disorder
Luciano et al. (2013) [77]	GWAS (Population)	Quantitative language across population	Language: Population study, low language determined based on non-word repetition tasks Cognition: -	-
Eicher et al. (2013) [78]	GWAS	Language Impairment (+/− RD)	Language: z-score ≤ −1 on ≥2 of 3 language tasks (phoneme deletion, verbal comprehension, non-word repetition) Cognition: IQ ≥ 76	-
Gialluisi et al. (2014) [81]	GWAS	Language Impairment (+/− RD)	Language: 3 cohorts with varied inclusion criteria: *1:* SLIC * CELF-R expressive or receptive SS ≥ 1.5SD below normative mean; *2*: UK Reading Disability: diagnosis RD; *3*: Colorado Learning Disabilities Research Centre: 2 datasets, one recruited on basis of diagnosis of RD, one on diagnosis of ADHD. Language SS≥3SD sample mean Cognition: *1*: * Performance IQ ≥ 80; *2:* Reading IQ discrepancy and/or *IQ > 90; 3*: FSIQ ≥ 70	*1*: * MZ twinning, chronic illness requiring multiple hospital visits or admissions, deafness, an ICD-10/DSM-IV diagnosis of childhood autism, EAL, care provision by local authorities, and known neurological disorders; *2: −3:* If ≥3 SS were ≥3SD from mean
Harlaar et al. (2014) [82]	GWAS (Population)	Quantitative language across population	Language: Population study, low language determined using receptive language measures included in the cognitive test battery Cognition: -	-
St Pourcain et al. (2014) [80]	GWAS	Quantitative language across population	Language: Population study, low language determined using MCDI Cognition: -	-
Nudel et al. (2014) [79]	GWAS	SLI	Language: * CELF-R expressive or receptive SS ≥ 1.5SD below normative mean Cognition: * Performance IQ ≥ 80	* MZ twinning, chronic illness requiring multiple hospital visits or admissions, deafness, an ICD-10/DSM-IV diagnosis of childhood autism, EAL, care provision by local authorities, and known neurological disorders
Evans et al. (2015) [86]	Linkage study	Poor language	Language: Recruited from a longitudinal language study. Overall language score calculated based on 3 composite language scores across general language, vocabulary and sentence use Cognition: Performance IQ > 70	-
Kornilov et al. (2016) [54]	GWAS Isolated population ~400	DLD	Language: Impairment (z-score < −1) on ≥2 quantitative phenotypes obtained via analysis of semi-structured speech samples Cognition: -	Children attending specialist education settings
Devanna et al. (2018) [111]	Sequencing study	SLI	Language: * CELF-R expressive or receptive SS ≥ 1.5SD below normative mean Cognition: * Performance IQ ≥ 80	* MZ twinning, chronic illness requiring multiple hospital visits or admissions, deafness, an ICD-10/DSM-IV diagnosis of childhood autism, EAL, care provision by local authorities, and known neurological disorders
Chen et al. (2017) [34]	Sequencing study (SLIC Cohort)	Severe SLI	Language: * CELF-R expressive or receptive SS ≥ 1.5SD below normative mean Cognition: * Performance IQ ≥ 80	* MZ twinning, chronic illness requiring multiple hospital visits or admissions, deafness, an ICD-10/DSM-IV diagnosis of childhood autism, EAL, care provision by local authorities, and known neurological disorders
Andres et al. (2019) [87]	Linkage study and homozygositymapping	SLI	Language: Peabody Picture Vocabulary Test (PPVT) fourth edition (PPVT-4) standard score of ≤80 Teacher report of SLI Cognition: -	Known developmental disabilities, HI, known neurological disorders
Andres et al. (2022) [37]	Sequencing study	SLI	Language: ≥1.0SD below mean on age-appropriate language test battery Cognition: Nonverbal-IQ > 85 on Columbia Mental Maturity Scale (age 3.6 to 6.11) or >85 on Wechsler Intelligence Test for Children or >85 Wechsler Intelligence Test for Adults	Known developmental disabilities, HI, known neurological disorders

* = study used SLIC cohort criteria; ADHD = attention-deficit/hyperactivity disorder; ASD = autism spectrum disorder; EAL = English as an additional language; HI = hearing impairment; IQ = intelligence quotient; MCDI = MacArthur–Bates communicative development inventory; RD = reading disorder; SD = standard deviation; SNHL = sensorineural hearing loss; SS = standard score.

## Data Availability

Not applicable.

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
