# Peer review of "The Genetic and Molecular Basis of Developmental Language Disorder: A Review"

_children, 2022, doi:10.3390/children9050586_

Round 1
Reviewer 1 Report
Thank you for your review paper. I enjoyed reading it. I did notice a few missing items that need to be addressed.
- Line 402 references a section on pathways, however there is no section describing pathway results. Either you decided against this section and missed this reference during your edits or somehow the section was not included in the draft you submitted.
- Lines 72- 74 you mention review past NVIQ standards. I would recommend you read/reference GALLINET & SPALDING, 2014 (meta analysis of NVIQ in DLD) to better describe the range of NVIQ that past research samples have had. The range is quite staggering and shows that not even the historical SLI literature stuck to the 85 SS "rule."
- Line 70 you could reference Tomblin et al 1996 (A system for the diagnosis of SLI in kindergarten children) where they describe their diagnostic rules and how these rules were developed. Many researchers in the USA cite that paper when justifying their SLI criteria, including the NIVQ criteria. Additionally, Stark & Tallal (1981) is frequently cited for NVIQ cutpoint justification.
- Lines 172 - 206, I think you are missing work here by Nicole Landi & Tracy Centanni on BDNF. They are primarily focused on reading, but you do bring up that reading is related to language. There is also additional work by Nicole Landi on SETBP1. See Landi & Perdue 2019 for a review of genes (including BDNF & SETBP1, and others) associated with language and reading from the neurogenetics work.
- Table 2 - possibly missing GWAS/GWAS like studies (depends on if you how you are defining language disorder):
Carrion-Castillo et al 2020 language and reading skill
Gialluisi et al (2020) developmental dyslexia
Shapland, Verhorf et al (2021) multivariate GWAS language, literacy, and working memory
Davis et al (2014) reading & math
Field et al (2013) - dyslexia
Verhoef et al (2020) & (2021) - early childhood vocabulary, genome wide complex trait analysis - Lines 367 - 372 gene X gene studies (dyslexia/reading):
Trezzi et al (2017)
Mascheretti S, Bureau A, Trezzi V, et al. An assessment of gene-by-gene interactions as a tool to unfold missing heritability in dyslexia. Human genetics 2015;134:749–60. doi:10.1007/s00439-015-1555-4
Powers NR, Eicher JD, Miller LL, et al. The regulatory element READ1 epistatically influences reading and language, with both deleterious and protective alleles. Journal of Medical Genetics 2016;53:163–71. doi:10.1136/jmedgenet-2015-103418 - Line 374 gene X environment (dyslexia) studies:
Mascheretti S, Bureau A, Battaglia M, et al. An assessment of gene-by-environment interactions in developmental dyslexia-related phenotypes. Genes, Brain and Behavior 2013;12:47–55. doi:10.1111/gbb.12000
McGrath et al (2007)
Reviewer 2 Report
I enjoyed reading this timely and clear overview of the current state of knowledge on the genetics of developmental language disorder, and think it will be a really useful addition to the literature. I felt the organisation and coverage of the review was appropriate and easy to follow. I have just one significant (but easily addressed) concern regarding content, and a small number of minor suggestions to further enhance clarity.
1) My one substantive concern is with respect to the characterisation of the quantitiatve genetic (mainly twin) research on DLD, which I don't think is an accurate reflection of the field. I understand that this is not the core body of research being reviewed here, and I agree a short section is all that's needed, but it does motivate what comes next in terms of the molecular genetic work. In short, you present the findings as consistently supporting substantial heritability for DLD, and cite Hayiou-Thomas (2005) as reporting heritability of DLD/SLI at 75%. In fact, heritability of SLI in that paper is reported as 18%. A follow-up study by Bishop & Hayiou-Thomas (2008) clearly demonstrated an enormous difference in heritability estimates based on 'clinically' vs 'psychometrically/epidemiologically' ascertained samples of SLI/DLD, where heritability was exteremely high for the clinical group (.97), but zero for the epidemiological group. What differentiated these two groups was the presence of speech difficulties in the clinical group. That is, it was the speech difficulties that appeared to be driving the heritability (in the TEDS sample, but also in the small number of twin studies up until that point, reviewed in the 2008 paper), while DLD in young children who did not also have speech difficulties appeared to be largely environmental in origin.
I agree there does seem to be some evidence that more severe language disorder is more heritable (e.g. DeThorne et al., 2005, JSLHR), and also that the picture may change with age: low verbal skills at 12 seem to be more heritable (Haworth et al., 2009 JCPP; Dale et al., 2018), consistent with a broader increase in heritability for language (Hayiou-Thomas et al., 2012, Dev Sci; Verhoef et al., 2021 JCPP).
In sum: I don't think you need a comprehensive review of this literature, but I think you need to get across that the case for high heritability of DLD is not at all clear-cut, and this actually makes it quite different from autism, ADHD and dyslexia where there is a lot of convergent and consistent evidence of strong genetic effects.
2) Perhaps not coincidentally, quite a lot of what you present in terms of genetic findings is actually about CAS (under monogenic disorders), and dyslexia (under GWAS), rather than DLD. Although this is very much an accurate account of the field, I think that the framing of the review may need to be altered a bit to reflect that more clearly.
3) Table 1 woud be easier to read organised by phenotype. I also wondered whether there was an organising principle that would make Table 3 easier to digest.
4) In general I found the explanations of genetic methodology to be clear and accessible to a broad audience, but there were a couple of places where I thought they could be expanded/clarified a little more: i) in the discussion of monogenic disorders, I would spell out that there are many variants within a single gene, ii) define 'throughput methodology' (p 6), iii) explain how polygenic risk scores are derived (pp 8-9).
5) CNVs, penultimate para on p. 5 who say' mapping of deletions....has been extended to DLD' but it's not clear if any of the work discused in that paragraph relates directly to DLD.
6) There are some minor ommissions/errors/sentence fragments which will be easy to tidy up. (I haven't made a comprehensive list, but e.g. p. 3 'monogenic disorders have traditionally been performed within families'/ p. 5, line 5, 'SLI' followed by 'DLD'; p. 7 about halfway down 'Andres et al. (2019) used linkage...' but the sentence leading up to it implies you're now talking about GWAS).
